

# Methane profiles from GOSAT thermal infrared spectra

Arno de Lange[1] and Jochen Landgraf[1]

[1]SRON Netherlands Institute for Space Research, Utrecht, The Netherlands

*Correspondence to:* Arno de Lange (A.de.Lange@sron.nl)

**Abstract.** This paper discusses the retrieval of atmospheric methane profiles from the thermal infrared band of the Japanese Greenhouse Gases Observing Satellite (GOSAT) between 1210 and 1310 cm$^{-1}$, using the RemoTeC analysis software. Approximately one degree of information on the vertical methane distribution is inferred from the measurements with the main sensitivity at about 9 km altitude but little sensitivity to methane in the lower troposphere. For verification, we compare the

GOSAT methane abundance at measurement sites of the Total Carbon Column Observing Network (TCCON) to methane profiles provided by the Monitoring Atmospheric Composition and Climate (MACC) model fields scaled to the total column observations at the sites. Without any radiometric corrections of GOSAT observations, differences between both data sets can be as large as 10%. To mitigate these differences, we developed a correction scheme using a principal component analysis of spectral fit residuals and airborne observations of methane during the HIAPER Pole-to-Pole Observations (HIPPO) campaign

II and III. When the correction scheme is applied, the bias in the methane profile can be reduced to less than 2% over the whole altitude range with respect to MACC model methane fields. Furthermore, we show that, with this correction, the retrievals result in smooth methane fields over land and ocean crossings and no differences are to be discerned between daytime and nighttime measurements. Finally, a cloud filter is developed for the nighttime and ocean measurements. This filter is rooted in the GOSAT-TIR measurements and is consistent with the cloud filter based on the GOSAT-SWIR measurements, despite the

fact that the TIR-filter is less stringent.

## 1 Introduction

Methane ($CH_4$) is, after carbon dioxide ($CO_2$), the strongest anthropogenic greenhouse gas with an estimated total radiative forcing of 0.97 W/m$^2$ for 2011 with respect to the pre-industrial levels of 1750 (Myhre et al., 2013). The forcing per molecule is ≈100 times stronger than that of carbon dioxide, but the abundance is ≈200 times lower. The current relative increase

with respect to pre-industrial background levels of 1750 is 150% for methane, as opposed to 40% for carbon dioxide. Natural methane sources are anaerobic environments where micro-organisms convert organic material into methane. Examples are wetlands including swamps, boreal marshes, and tundras, but also lakes and oceans are natural methane sources. In line with these natural processes are the cultivation of rice paddies and cattle, both anthropogenic sources. Other anthropogenic sources include the burning of organic material (biomass burning and waste burning), and gas losses in the fossil fuel industry (Myhre

et al., 2013). For climate monitoring and prediction, it is essential to measure $CH_4$ on a global scale and information on its vertical distribution may help to disentangle signals due to methane surface emissions and long-range transport.



Satellite nadir measurements of $CH_4$ in the thermal infrared (TIR) represent an important element of a climate observing system because of the pronounced methane sensitivity of the measurements in the upper troposphere. Therefore, these measurements can aid in the decoupling of methane emissions and transport in inverse-modeling studies. Currently five nadir-viewing instruments in the thermal infrared are operational on satellite platforms; Atmospheric InfraRed Sounder (AIRS), aboard Aqua,

launched in 2002 (Xiong et al., 2008; Zou et al., 2016); Tropospheric Emission Spectrometer (TES), aboard AURA, launched in 2004 (Wecht et al., 2012; Worden et al., 2012, 2015); Infrared Atmospheric Sounding Interferometer (IASI), aboard Metop-A and Metop-B, launched in respectively 2006 and 2012 (Razavi et al., 2009; Crevoisier et al., 2013; Cressot et al., 2014; Siddans et al., 2016); Thermal and Near infrared Sensor for Carbon Observation - Fourier Transform Spectrometer (TANSO-FTS) aboard GOSAT, launched in 2009. Furthermore, GOSAT-2, the successor of the GOSAT satellite, will also be equipped

with two thermal infrared bands (together covering the same wavelength range as the one TIR band in GOSAT) and a new generation of IASI spectrometers (IASI-NG) will fly on three successive Metop-SG A satellites of the EUMETSAT Polar System of Second Generation (EPS-SG) in the 2021-2040 time frame.

In many studies on methane retrievals from thermal infrared observations of the above mentioned satellites, a bias in the methane product is observed. To address this bias, different approaches have been adopted. Worden et al. (2012) observes a

discrepancy between upper and lower tropospherical methane of $\approx 4\,\%$ in case of TES observations and mentions uncertainties in temperature, calibration inaccuracies and spectroscopy errors as the main causes. A $CH_4/N_2O$ proxy retrieval reduces this bias to $\approx 2.8\%$ but does not fully remove it. Siddans et al. (2016) also observes a bias of $\approx 4\%$ and scales the methane mixing ratios retrieved from IASI measurements. Furthermore, two additional scaling parameters are fitted for the mean residual of respectively nadir observations and at the outer edge of the swath, to account for variations in interfering water vapor and scan

mirror errors. The resulting methane product is within 2% of HIPPO over the full altitude range. Also Crevoisier et al. (2013) applies a radiometric correction based on mean residuals in case of IASI measurements. Finally, von Clarmann et al. (2009) mentions that 8 micro-windows are carefully selected for the MIPAS limb retrievals to reduce the known high bias of $CH_4$.

The current study focuses on the retrieval of methane profiles from GOSAT observation in the thermal infrared band 1210–1310 $\mathrm{cm}^{-1}$. Previous work by Saitoh et al. (2012); Holl et al. (2016) presented first results of GOSAT-TIR retrievals of methane,

where the measurements are radiometrically corrected using the approach of Saitoh et al. (2009). Spectral residuals in GOSAT are assessed with measurements of buoys and are subsequently accounted for in the retrievals. Here, the degree of signal for the retrieval is significantly lower than 1. Moreover, Zou et al. (2016) compared the retrieval of methane profiles from GOSAT and IASI thermal infrared measurements and observed a prevalent bias at 9 $\mathrm{km}$ altitude of about 3% between both satellite retrievals.

In this study, we apply the RemoTeC retrieval tool to analyse the GOSAT-TIR measurements with a degree of freedom for signal (DOFS) $\approx 1$ and verify the retrieval results with profiles of the Monitoring of Atmospheric Composition and Climate (MACC) project scaled to the total column measurements of the Total Carbon Column Network (TCCON) at ground-based measurements sites. To mitigate the observed significant biases, we developed a sophisticated correction scheme based on a principal component analysis of spectral residuals using HIPPO (HIAPER Pole-to-Pole Observations) data as an estimate for

the atmospheric state. This correction scheme improves significantly our validation. Moreover, we achieve the consistency



of daytime and nighttime measurements as well as continuity of methane for land–sea crossings where the measurement sensitivity to methane in the lower and middle atmosphere changes substantially.

The article is structured as follows: in Section 2 the GOSAT-TIR measurements are introduced, Section 3 introduces the Tikhonov regularisation scheme to invert the measurements. Finally, Section 4 presents the bias correction scheme and its
effect on the retrievals is demonstrated.

## 2   GOSAT

The Japanese satellite GOSAT (Greenhouse gases Observing SATellite) was launched in 2009 and is the world's first satellite fully dedicated to the monitoring of the two most important greenhouse gases — carbon dioxide and methane. Its main instrument is the TANSO-FTS Fourier Transform spectrometer covering four wavelength bands; the oxygen A-band ($\approx$
$13000$ cm$^{-1}$), two bands in the shortwave infrared regime ($\approx 5000$ cm$^{-1}$ and $\approx 6200$ cm$^{-1}$), and a band in the thermal infrared wavelength range ($\approx 600$–$1600$ cm$^{-1}$). This study focuses only on the retrieval of methane for the thermal infrared (TIR) band, employing the spectral window $1210$–$1310$ cm$^{-1}$. TANSO-FTS has an instantaneous field of view of $\approx 10$ km allowing for high spatial resolution measurements with at the same time a sparse spatial sampling. Here, the distance between two consecutive measurements is up to several $100$ km.

Over the coarse of the mission, JAXA (Japan Aerospace Exploration Agency) has released several level 1B data versions, and in this study v16x160 of L1B data has been used. For the TIR band, this version contains important updates with respect to previous versions (such as updated radiometric correction parameters, polarization effects, and reference blackbody emissivity) and is virtually identical to the most recent version v201202. For further details on the instrument, its calibration, and performance, we refer to Kuze et al. (2009, 2014, 2016).

## 20   3   Retrieval

To infer methane profiles from GOSAT-TIR measurements, a forward model $\boldsymbol{F}$ is needed, that simulates the radiance measurement $\boldsymbol{r}$ as function of the atmospheric state vector,

$$\boldsymbol{r} = \boldsymbol{F}(\boldsymbol{x}, \boldsymbol{b}) + \boldsymbol{e_y}, \tag{1}$$

where $\boldsymbol{e_y}$ comprises forward model error and instrument error including the measurement noise. The state vector $\boldsymbol{x}$ contains
all parameters to be retrieved from the measurement and consists of the methane profile (defined over 12 layers at equidistant pressure levels), the skin temperature, a spectral shift, and four scaling parameters for the total columns of $H_2O$, HDO, $N_2O$, and an effective total $H_2O$ column to calculate the water-continuum independently from the water vapour absorption lines. The forward model parameter $\boldsymbol{b}$ symbolises all model parameters that require prior knowledge, such as instrument parameters, atmospheric pressure and temperature profiles.





### 3.1 Forward model

To simulate line-by-line radiance spectra at the top of the model atmosphere, we account for the Planck radiation of the Earth surface and its atmosphere as radiation source and ignore the solar contribution to the spectrum analogous to e.g. Wassmann et al. (2011). For a non-scattering atmosphere, the down-welling radiation is reflected by the Earth surface as-
suming isotropic Lambertian reflection. The down-welling is calculated by the means of a 4-point Gaussian quadrature. The wavelength-dependent emission by the Earth's surface is a function of surface temperature and the surface emissivity. Here, the surface reflection of down-welling atmospheric radiation is governed by the emissivity, following Kirchhoff's law. We determine the initial surface emissivity over land with the High Spectral Resolution Algorithm developed by Borbas (Borbas et al., 2007; Borbas and Ruston, 2010) using the University of Wisconsin Baseline Fit (UW-BF) Emissivity Database (Seemann et al.,
2008) as input. For water surfaces, we use the IRSSE model by van Delst and Wu (2000) to calculate the surface emissivity in RemoTeC. This model is an update of Wu and Smith (1997) and the calculated emissivity is a function of sea-roughness as determined by the wind speed, viewing angle, and wavelength of the radiation. Furthermore, we consider atmospheric absorption by $H_2O$, $CH_4$, $N_2O$, and $CO_2$ from the HITRAN 2008 database (Rothman et al., 2009) and describe atmospheric continuum absorption using the model by Mlawer et al. (2012) to account for broad-band contributions from water, carbon
dioxide, oxygen, nitrogen, and ozone. Here the continuum contribution by water is calculated separately, including the foreign and self continuum. The surface temperature and wind speed as well as the water vapour and temperature profile of the atmosphere, needed to initialise the retrieval, are taken from European Centre for Medium-Range Weather Forecast (ECMWF) ERA interim data set (Dee et al., 2011). The $CH_4$ and $N_2O$ profiles are adapted from MACC-II (Bergamaschi and Alexe, 2014), and $CO_2$ from CarbonTracker CT2013 (Peters et al., 2007; CarbonTracker website).
Finally the line-by-line spectra are degraded to the spectral resolution of the sensor using TANSO-FTS spectral response. Numerically, the forward model is implemented in the RemoTeC retrieval tool (Butz et al., 2011; Schepers et al., 2012) to benefit from the overall processing properties of this framework.

### 3.2 Inversion

The goal of a retrieval is to determine the atmospheric state vector $x$ in Eq. (1) by inverting the forward model $F$. The RemoTeC
inversion module is described in detail by Butz et al. (2011) and in this section we summarise the inversion aspects that are relevant for this study.

Since the forward model $F$ is generally not linear in the state vector $x$, Eq. (1) is inverted with a Gauss–Newton iteration scheme. This means that $F$ is linearised in each iteration step by a Taylor expansion around the solution of the previous step, starting with a first guess state vector $x_0$. Equation (1) can be rewritten as

$$y = \mathbf{K}x + e_y, \tag{2}$$

where $y = r - F(x_{i-1}) + \mathbf{K}x_{i-1}$ is the so-called measurement vector with $x_{i-1}$ the state vector of the previous iteration step and $\mathbf{K}$ is the Jacobian matrix.





To infer a vertical methane profile from GOSAT-TIR measurements represents an ill-posed problem requiring regularisation. Several techniques have been developed to solve such problems and in this study we employ 0-th order Tikhonov regularisation (Tikhonov, 1963; Phillips, 1962; Twomey, 1963):

$$x_\gamma = \min_x \left( \|(\mathbf{K}x - y)\|^2 + \gamma^2 \|x\|^2 \right), \tag{3}$$

where $x_\gamma$ is the solution vector of the minimisation problem, $\|(\mathbf{K}x - y)\|^2$ is the least squares norm and $\|x\|^2$ is the side constraint. $\gamma$ is the regularisation parameter and has to be carefully chosen to balance the two contributions of the cost function. In this study, we employ the L-curve paradigm to find the appropriate value for the regularisation parameter (Hansen, 1992), which is discussed extensively in the literature (O.P. Hasekamp, 2001; Steck, 2002; Butz et al., 2011). The solution of Eq. (3) is

$x_\gamma = \mathbf{D}y, \tag{4}$

where

$$\mathbf{D} = \left( \mathbf{K}^{\mathrm{T}}\mathbf{K} + \gamma^2\mathbf{I} \right)^{-1} \mathbf{K}^{\mathrm{T}}\mathbf{S}_y^{-\frac{1}{2}}, \tag{5}$$

is the pseudo-inverse of $\mathbf{K}$ or the contribution matrix, $\mathbf{S}_y$ the measurement noise covariance matrix and $\mathbf{I}$ represents the unity matrix. Moreover, the retrieval noise covariance matrix is given by

$\mathbf{S}_x = \mathbf{D}\mathbf{S}_y\mathbf{D}^{\mathrm{T}}, \tag{6}$

where $\mathbf{S}_y$ is measurement noise covariance.

    The retrieved vector $x_\gamma$ is a weighted average of the true atmospheric state vector $x_{\mathrm{true}}$

$$x_\gamma = \mathbf{A}x_{\mathrm{true}} + e_x, \tag{7}$$

due to the required regularisation. Here $\mathbf{A} = \mathbf{DK}$ is the averaging kernel and $e_x = \mathbf{D}e_y$ is the error in the state vector caused
by measurement errors. Effectively, the averaging kernel degrades the true profile to the vertical resolution of the retrieval and also defines the null-space contribution of the true state vector $x_{\mathrm{true}}$, namely

$$x_{\mathrm{null}} = \left( \mathbf{I} - \mathbf{A} \right) x_{\mathrm{true}}, \tag{8}$$

comprising the contribution of the state vector, that cannot be inferred from the measurement.

    Figure 1 shows a typical averaging kernel for GOSAT-TIR retrievals. It indicates a peak around 9 km, and the retrieval
sensitivity drops quickly for altitudes close to surface. This loss in sensitivity to methane concentrations at low altitudes can be understood as follows. Line features only show up in the spectrum if the local photon field is not at equilibrium with the atmospheric radiance corresponding to the Planck curve governed by the ambient temperature. In general, the skin temperature of the Earth is similar to the temperature of the lowest layers of the atmosphere, and the radiance emitted by the Earth's surface



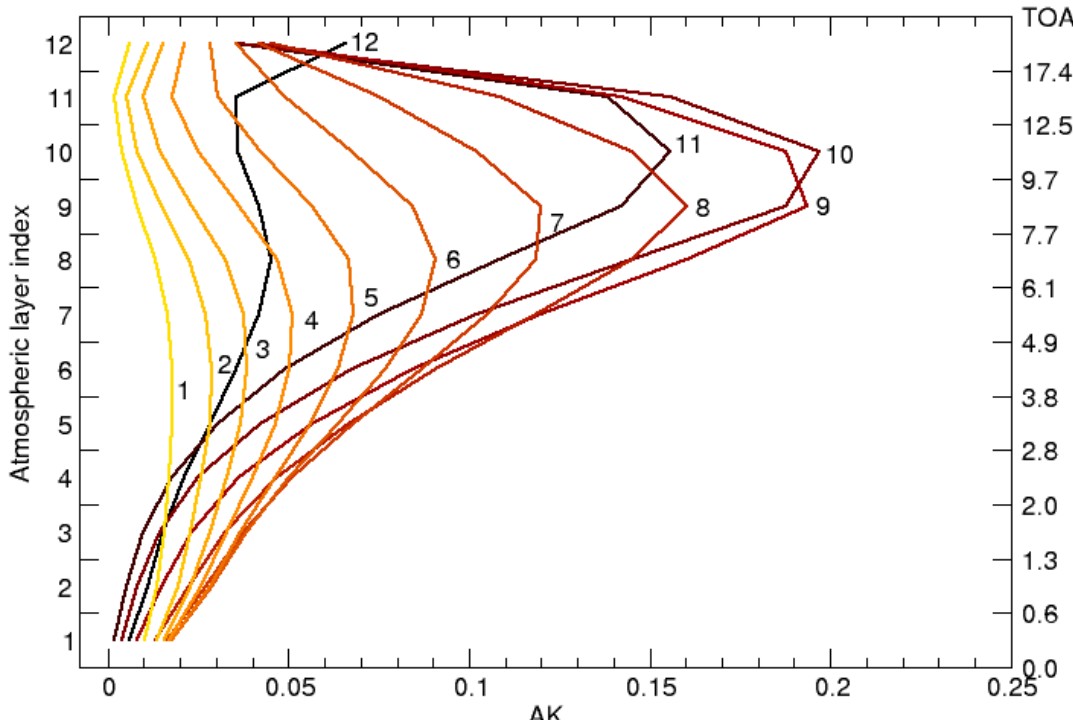

**Figure 1.** A typical averaging kernels for GOSAT-TIR daytime measurements over land. The colour coding is such that dark colours correspond to the averaging kernels at high altitudes and the bright colours to low altitudes. The numbers next to the curves refer to the corresponding atmospheric layer index as is indicated on the left side of the graph.

is in accordance with the radiances by those atmospheric levels. So, molecular line features are only weakly imprinted on the recorded spectrum, and hence, the sensitivity is very limited.

The degree of freedom for signal (DOFS) is defined by

$$\mathrm{DOFS} = \mathrm{Tr}\,\mathbf{A}, \tag{9}$$

5    and can be interpreted as the amount of independent pieces of information that can be retrieved from the measurement. In the case of GOSAT-TIR retrievals the DOFS is $\approx 1.0$ for scenes with a limited temperature contrast (nighttime measurements or daytime measurements over the ocean) and $\approx 1.1$ for daytime measurements over land with on average a slightly higher temperature contrast.





When one considers the retrieved profile $x_\gamma$ in Eq. (3) as the methane data product, one has to account for the reduced retrieval sensitivity given by the averaging kernel. Alternatively, one may consider the vertical profile after adding an estimate of the null-space contribution to $x_\gamma$, namely

$$x_{\mathrm{CH}_4} = x_\gamma + (\mathbf{I} - \mathbf{A})\, x_{\mathrm{apr}}, \tag{10}$$

with an a priori estimate of the true profile $x_{\mathrm{apr}}$ coming from e.g. an independent measurement or a chemical transport model forecast. For the purpose of our study, it is also valuable to calculate the total methane column $c_{\mathrm{CH}_4}$ from Eq. (10), namely

$$c_{\mathrm{CH}_4} = \mathbf{C} x_{\mathrm{CH}_4}, \tag{11}$$

where $\mathbf{C}$ is the column operator, effectively summing all partial columns in $x_{\mathrm{CH}_4}$. This implies that $x_{\mathrm{CH}_4}$ is given in column units, e.g. $\mathrm{cm}^{-2}$. To indicate the actual retrieval sensitivity the so-called column averaging kernel is a useful quantity, defined by the averaging kernel $\mathbf{A}$ and column operator $\mathbf{C}$ (Borsdorff et al., 2014)

$$A_{\mathrm{col}} = \mathbf{C} \mathbf{A}, \tag{12}$$

as well as the corresponding retrieval error variance

$$s_{\mathrm{CH}_4} = \mathbf{C} \mathbf{S}_x \mathbf{C}^{\mathrm{T}}. \tag{13}$$

A typical column averaging kernel for land and ocean scenes at daytime and nighttime is depicted in Fig. 2. It shows a clear drop-off in sensitivity towards the lower layers of the atmosphere and also the enhanced sensitivity to the lower layers for daytime measurements over land scenes can be discerned. This enhancement is because of the temperature contrast, which is larger for daytime than for the nighttime measurements and larger for measurements over land than over the ocean.

A pirori knowledge on methane has a significant impact on the profile $x_{\mathrm{CH}_4}$, which is indicated by the fact that the null-space contribution of the integrated methane column is typically in the order of 30%. Hence, the reference profile must be of sufficient quality not to govern the uncertainty in $x_{\mathrm{CH}_4}$.

### 3.3 Validation approach

Ideally, the validation of the retrieved GOSAT methane profile relies on independent validation measurements of the vertical methane distributions. A promising data source of validation is given by AirCore balloon soundings reaching a height up to 30 km (Karion et al., 2010). However, the development of an extended observation framework is still on-going and to this day the number of usable soundings is very limited causing poor statistics for the validation. Therefore, we decided not to consider these observations in our study. Another validation possibility is to use airborne measurements within the HIPPO (HIaper Pole to Pole Observations) project conducted in the years 2009–2011. Here methane profiles are measured up to $\approx 13$ km. In our study, however, these measurements are used to radiometrically correct the GOSAT measurements, as will be shown in Section 4, and can therefore not be used for validation purposes. Finally, during ascent and decent of commercial airlines the methane distribution is measured in-situ close to airports up to typical flight heights of 10 km. Two examples of





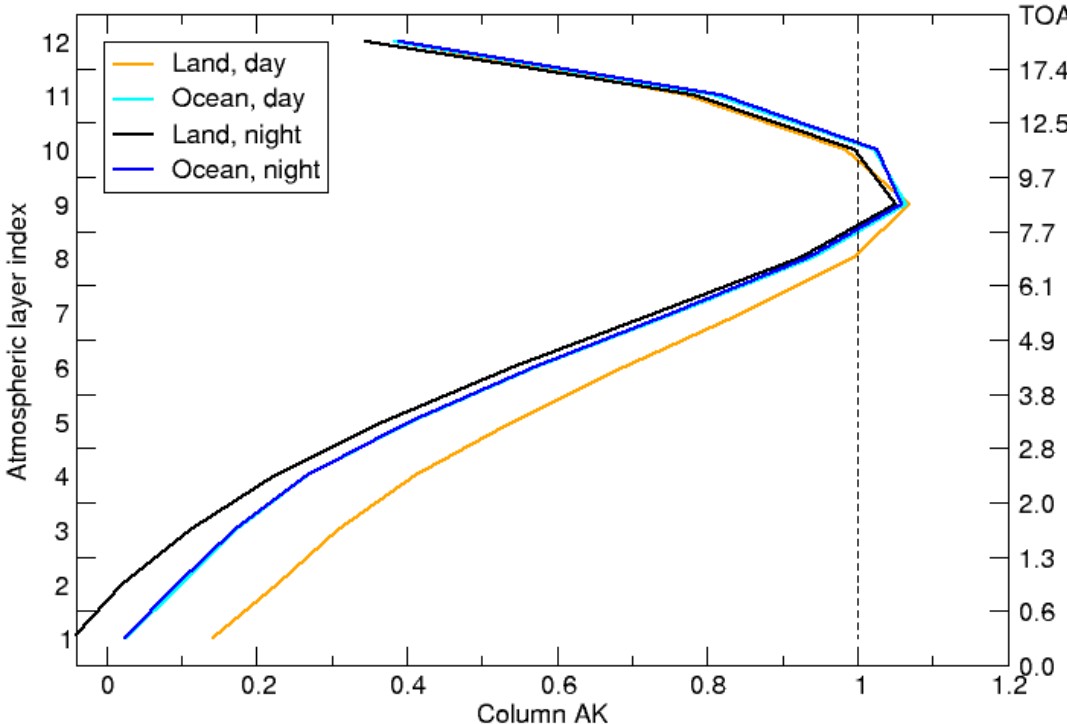

**Figure 2.** The average of all column averaging kernels in this study for different scenes and measurement times. Daytime and nighttime measurements over land are depicted in respectively orange and black curves and over the ocean in dark blue and cyan curves.

such measurement frameworks are CONTRAIL (Comprehensive Observation Network for Trace gases by Airliner) (Machida et al., 2008; Inoue et al., 2014) and CARIBIC (Civil Aircraft for the Regular Investigation of the atmosphere Based on an Instrument Container) (Brenninkmeijer et al., 2007). Obviously, these measurements are limited in altitude to typically 10 km and therefore lack an important part of the profile to which the GOSAT-TIR measurement is sensitive to. All in all, high-quality measurements of methane profiles are sparse and due to the required co-location between GOSAT measurements and the available reference measurements, we estimate the number of validated profiles to be too limited for a validation of our product.

A common approach to validate methane total column retrieval from shortwave infrared measurements is to compare the retrieved column to co-located ground-based measurements of the Total Carbon Column Observing Network (TCCON) (Wunch et al., 2011). Here, both the ground-based and satellite observations show homogeneous methane retrieval sensitivity over all





atmospheric altitudes, leading to highly accurate estimates of the total column of methane rather than a profile. Because of the lack of other validation measurements, we decided to use these data for our product validation. In first instance, we derive a methane profile employing the MACC-II repository (Bergamaschi and Alexe, 2014), which delivers global methane fields on a daily basis, to extract an a priori estimate $\boldsymbol{x}_{\mathrm{apr}}$. This estimate is subsequently scaled to the total columns of co-located

TCCON measurements. The profile is used to derive the total methane column as described by Eq. (11) and so the comparison with TCCON total columns ensures the same null-space contribution in the GOSAT TIR and the validation estimate of the total column. Moreover, the MACC/TCCON profile can be used to compare to the methane profile $\boldsymbol{x}_{\gamma}$ in Eq. (7) and $\boldsymbol{x}_{\mathrm{CH_4}}$ in Eq. (10).

## 3.4  CH$_4$ retrievals

We start our validation analysis comparing the retrieved total methane columns from GOSAT-TIR measurements over TCCON stations with the corresponding measurements at the site. To minimise the interfering effect of scattering by clouds, hazes, cirrus, and/or aerosols, in this study we pertain to clear-sky conditions. This implies that a cloud-filter needs to be employed. Although, within the RemoTeC-framework, a well-tested cloud filter is available for daytime measurements over land based on SWIR and NIR spectra, there is no equivalent for the TIR spectrum. Particularly to filter nighttime measurements or mea-

surements over the ocean, one has to resort to a cloud filter rooted in the TIR spectra. Therefore, we make use of the fact that clouds generally change the effective light path by scattering and the retrieved columns differ from the actual columns. For cloud clearing of the data set, we consider the difference between the retrieved N$_2$O total column and the MACC N$_2$O total column. To account for an overall bias in the MACC N$_2$O columns, we define a dynamical mean $x$ of the N$_2$O column errors such that cumulative number of converged retrievals in the range $[x - 3\%, x + 3\%]$ are maximised. All data within this interval

are considered in the successive analysis to be cloud filtered. In Section 4.1 the cloud filter is investigated in more detail. Moreover, data are filtered using a stringent normed Pearson's chi-squared criterium of $\chi^2 < 3.0$ for the spectral fit quality. Figure 3 depicts the comparison between retrieved total methane columns from GOSAT-TIR measurements and co-located TCCON observations. Although, the GOSAT methane results capture the seasonal variation, they also clearly show a large and persistent bias of about 4.6%.

This bias shows little variation when compared to nine other TCCON sites (Bialystok, Bremen, Darwin, Lauder, Orleans, Park Falls, Reunion, Sodankyla, and Wollongong) as depicted in Figure 4. The error bars indicate the $1\sigma$ standard deviation of the difference between GOSAT and TCCON and correspond to a typical value of 2%. The propagation of the measurement noise into the retrieved total column amounts to retrieval error $s_{\mathrm{XCH_4}} \approx 0.8\%$ and explains only a part of spread. The average bias is $+4.6\%$, whereas the station-to-station variation in the bias is much smaller (0.4%). The accuracy of the total columns

from the MACC repository, used as prior in the retrieval product, is also estimated to be of the order of 2%. This estimation is based on the study in Landgraf et al. (2016) where methane fields from the TM5 model are compared against GOSAT-SWIR retrievals and it was found that on average the standard deviations are well within 1%, with sporadic outliers up to 3%. It is noted that the TM5 model runs were conducted with methane constraints only taken from the measurements of the NOAA-ESRL global monitoring network, whereas within the MACC repository also the GOSAT-SWIR measurements are





**Figure 3.** Total methane columns as a time series over Lamont. In black the GOSAT-TIR retrievals are shown. In respectively orange and cyan, the corresponding (non-scaled) MACC and TCCON total columns are shown.

taken as input. Therefore, we believe that the estimation of 2% on the accuracy of the total columns from the MACC repository is reasonable, even on the safe side. Since it typically contributes for 30% to the retrieved total column, it contributes with $\approx 0.6\%$ to the error budget. Finally, the precision and accuracy of the methane TCCON measurements are both estimated to be $< 0.3\%$ (TCCON Data Description website). Overall these relatively small error contributions implies that further sources

5  of uncertainties exist, e.g. radiometric calibration and forward model errors.

To better understand the induced errors, we compare the average GOSAT-TIR methane profile $x_{\mathrm{CH_4}}$ over the TCCON site Bialystok with the averaged MACC/TCCON profile in Fig. 5. The deviation between the two profiles peaks around 9 km, which corresponds to the altitude of maximum retrieval sensitivity. This suggests that the altitude dependent bias in this figure finds its origin in the altitude dependent sensitivity. The depicted difference is indicatory for all TCCON stations.





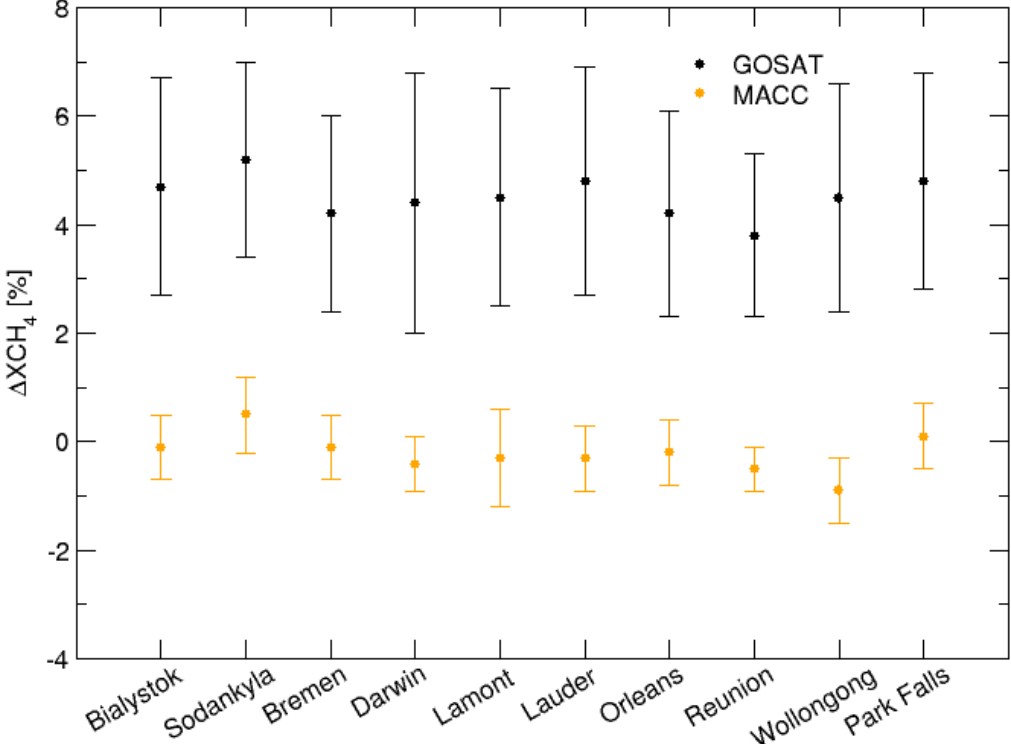

**Figure 4.** Average of the relative total methane column with respect to TCCON measurements for GOSAT-TIR retrievals (black) and the (non-scaled) MACC prior (orange). The bars indicate the spread in the data ensemble.

Therefore, it is insightful to compare the retrieved GOSAT-TIR profiles $x_\gamma$ in Eq. (4) with the smoothed MACC profile by applying the averaging kernel as indicated in Eq. (7). Figure 6 shows corresponding results for the TCCON site Bialystok. The bias on $x_\gamma$ is much more constant over altitude, but still shows some striking vertical features with biases peaking around 9 km with the highest sensitivity to methane, a negative lobe towards lower altitudes and a sharp drop-off for the upper layers of the

5    atmosphere.

To survey the retrieval performance at all TCCON sites mentioned in Table 1, Figure 7 shows the error histogram of the mean retrieval bias at 2 and 9 km. Overall we see a similar behaviour for all TCCON sites and the station-to-station bias variation is small ($\approx 0.6\%$). Moreover, we find an interesting variation of biases for different types of observations, where we distinguish between daytime and nighttime observations and land and ocean scenes. In Fig. 7, the bias at 9 km is systematically lower for

10    the daytime-land measurements than for the other three scenes, who are amongst themselves very comparable. At 2 km this




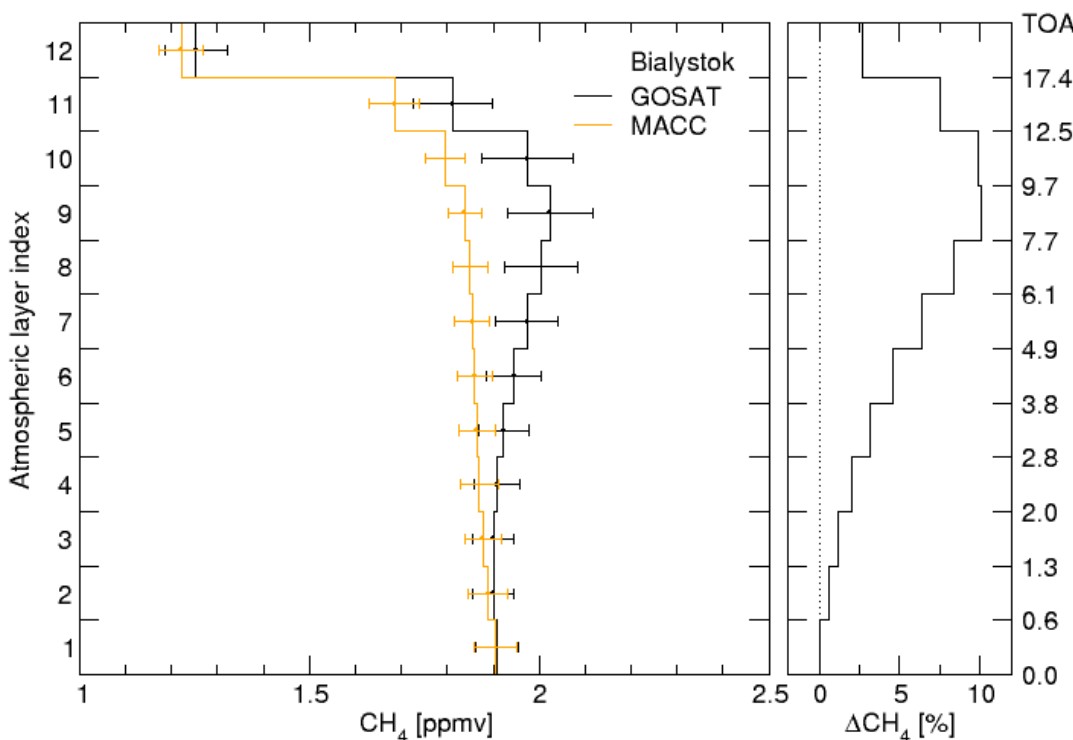

**Figure 5.** (left) Average methane profile retrieved from GOSAT-TIR spectra (black) over TCCON station Bialystok and from the MACC repository (orange) scaled such that the total column equals the corresponding TCCON total column measurement. The bars indicate the spread in the data ensemble. (right) The relative difference between the averaged retrieved GOSAT TIR and MACC methane profiles. It is noted that the two panels have different horizontal scales.





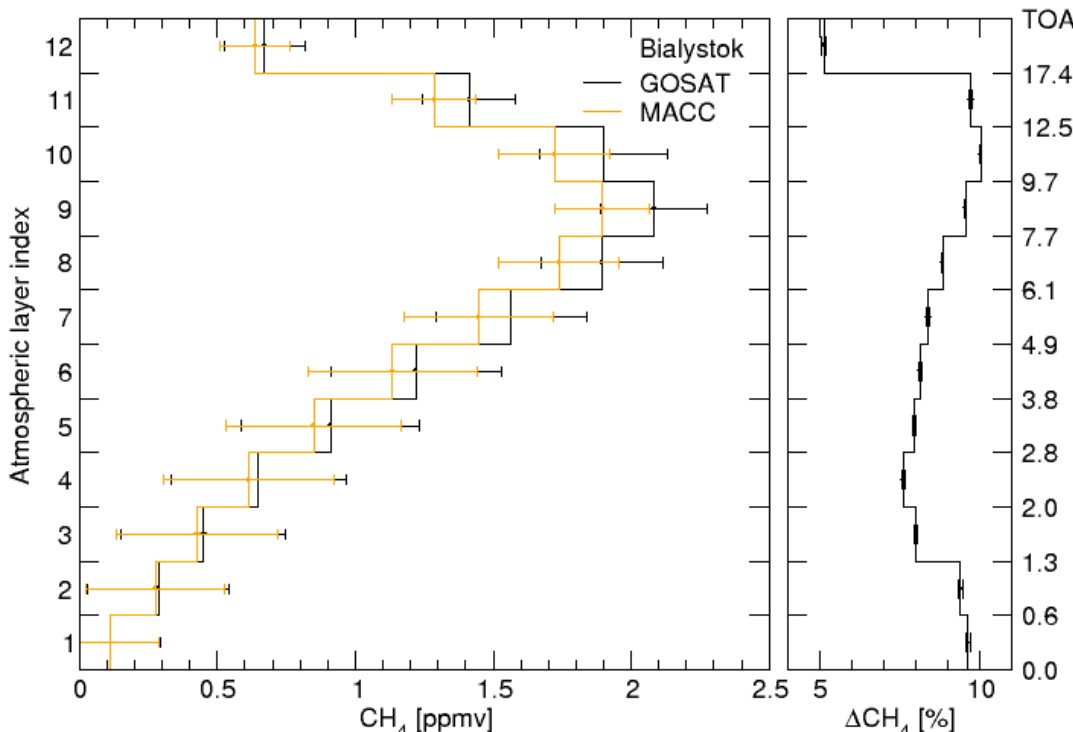

**Figure 6.** (left) Averaging kernel-smoothed profiles from GOSAT daytime measurements over land (black) and MACC (orange) at TCCON station Bialystok. The bars indicate the spread in the data ensemble. (right) The relative difference between the profiles. The bars pertain to the $1\sigma$ uncertainty of the averaged ratio, derived from the instrument noise propagation.

behaviour is reversed; daytime-land measurements are systematically higher. For the interpretation, we have to consider the different retrieval sensitivity as indicated in Fig. 2. During day over land, the thermal contrast in the lower atmosphere is larger than for the other three cases and therefore the retrieval sensitivity increases accordingly. This enhancement goes along with larger biases.

5    Overall we conclude that the biases in the retrieved $CH_4$ profiles are significant and requires a mitigation strategy. A straightforward scaling of the retrieved profile by a certain factor is not sufficient because it cannot account for the altitude dependent biases for both $x_\gamma$ and $x_{CH_4}$. Therefore, in the next section we will discuss a scheme to correct radiometrically the GOSAT-TIR measurements as part of the inversion.





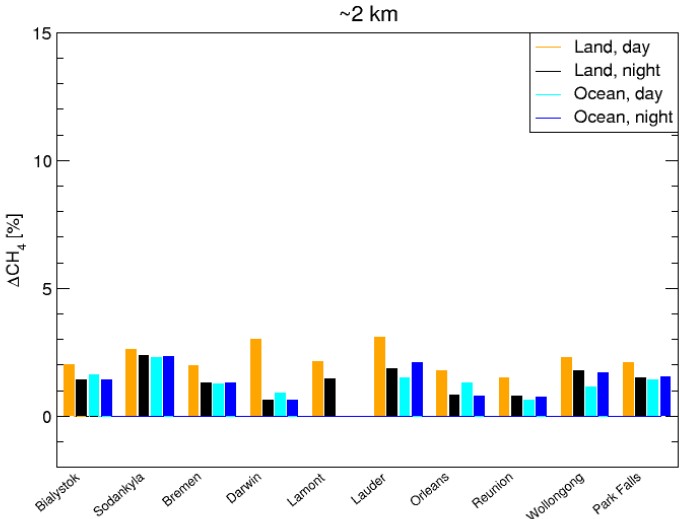

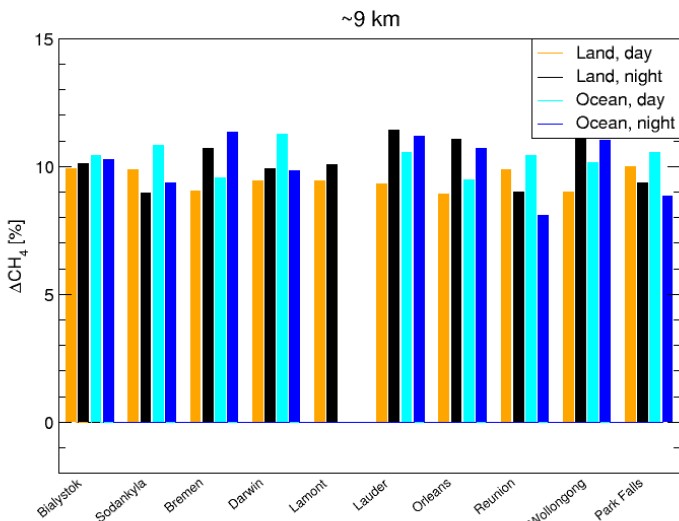

**Figure 7.** Histogram of the methane profile bias over ten different TCCON stations listed in Table 1, with the MACC/TCCON profiles as reference. The data are divided for different scenes; land and ocean scenes during daytime (respectively orange and cyan) and nighttime (respectively black and blue). In the left panel the deviation is shown for an altitude of $\approx 2$ km and in the right panel of $\approx 9$ km.



## 4 Bias correction scheme

Instead of correcting the biases at the level of geo-physical methane profiles, we consider an approach to quantify spectral features of radiometric biases of the GOSAT-TIR measurements. The observed methane bias finds its origin in the discrepancy $e_y$ between forward model $\boldsymbol{F}$ and measurement $\boldsymbol{r}$ in Eq. (1). Although we cannot distinguish between forward model errors

and instrumental errors, we can investigate the spectral properties of this discrepancy. Fixing the $CH_4$ and $N_2O$ profile to accurate a priori knowledge, we retrieve all other parameters of the state vector, i.e. the skin temperature, a spectral shift, the total columns of $H_2O$ and HDO, and an effective total $H_2O$ column to calculate the water-continuum independently from the water vapour absorption lines. Analysing spectral fit residuals guides us to identify spectral components of the radiometric bias, that interfere with the atmospheric methane absorption. For this purpose we used $CH_4$ and $N_2O$ data from the HIPPO

(HIAPER Pole-to-Pole Observations) campaigns II and III (held in, respectively, October 2009 and March 2010). The HIPPO data contains vertical profiles of many relevant species and atmospheric parameters, setting strong constraints on the estimated state of the atmosphere. Although most of the measurements are taken over the Pacific ocean, in both campaigns vertical profiles have also been recorded over Northern America, and, in the case of campaign III, also over New Zealand. Therefore, these campaigns seem to suit our need to include as many as possible different scenes to investigate systematics in spectral residuals.

With the co-location criteria ($\Delta\text{lat} = 5°$, $\Delta\text{lon} = 8°$, and $\Delta\text{t} = 2$ hrs), the amount of unique HIPPO-GOSAT measurement pairs is $\approx 300$.

Typically, the spectral residuals of this fit are very small, as can be seen in Figure 8. In the second panel, the noise level of a single measurement is indicated by the dashed line ($7 \times 10^{-8}$ W/m$^2$ sr cm$^{-1}$). The residual averaged over all co-located HIPPO-GOSAT pair is depicted in the third panel. Note that the spectral bias is less than 1% of the continuum level at

$1210$ cm$^{-1}$ for the depicted spectrum, but causes biases in the retrieved methane product up to 10% at 9 km altitude.

The comparison of the second and third panel of Fig. 8 indicates that most residuals average out for larger data sets. This may be due to random noise contributions but also spectral features which change from observation to observation in a non-random manner are suppressed by the averaging. Here the principal component analysis (PCA) provides an adequate mean to detect non-random contributions in the fit residuals. It is based on an eigenvalue analysis of the covariance matrix of the underlying

data set. The first principal component corresponds to the eigenvector with the largest possible variance, and for each succeeding component the variance degrades to lower values. By definition, the different principal components are uncorrelated.

Let $\mathbf{X}$ be the data matrix, consisting of 300 spectral residuals for all co-located HIPPO-GOSAT pairs, assuming that the mean residual is subtracted. Its covariance matrix $\mathbf{C}$ is then

$$\mathbf{C} = \mathbf{X}\mathbf{X}^{\mathrm{T}}/(n-1), \tag{14}$$

which is symmetric and the eigenvalue problem can therefore be written as

$$\mathbf{C} = \mathbf{V}\mathbf{L}\mathbf{V}^{\mathrm{T}}, \tag{15}$$

with $\mathbf{L}$ a diagonal matrix with the eigenvalues of $\mathbf{C}$ and $\mathbf{V}$ the set of eigenvectors. When $\mathbf{L}$ contains the eigenvalues in decreasing order, then the $i^{\mathrm{th}}$ principal component is the $i^{\mathrm{th}}$ column of $\mathbf{V}$.



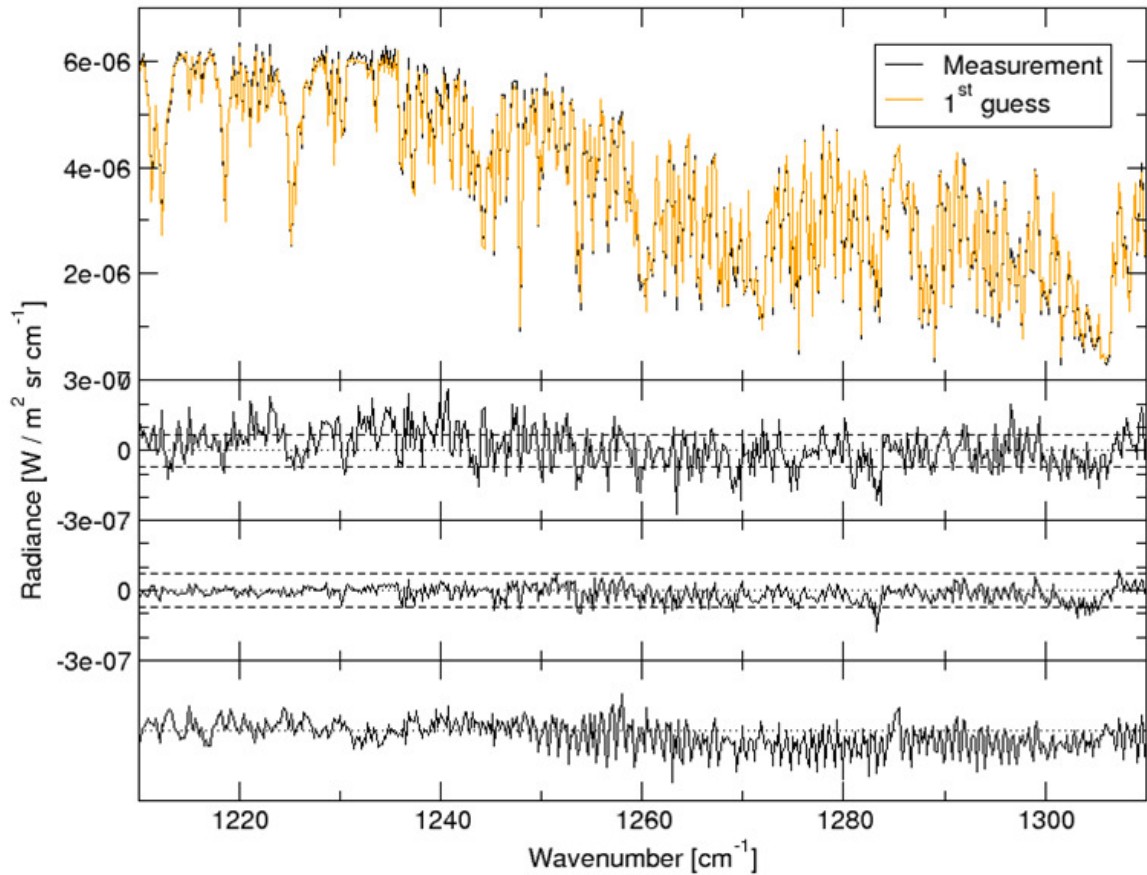

**Figure 8.** Comparison of GOSAT-TIR measurement with a forward model calculation based on methane profiles as measured during the HIPPO campaigns. The upper panel shows a single GOSAT measurement (black) and the forward calculation based on the co-located HIPPO measurement (orange), and in the second panel the residual is shown. The third panel depicts the average residual of all 300 GOSAT-HIPPO pairs used to centre all residuals in the principal component analysis. In the bottom panel the first principal component of this analysis is pictured.

The first principal component is shown in the fourth panel of Fig. 8. The strongest spectral features in this component show above 1250 $cm^{-1}$ and follow mostly $N_2O$ and $CH_4$ lines. In fact, this wavelength range corresponds to the part of the measurement where $N_2O$ and $CH_4$ are strongly interfering. Between 1210 $cm^{-1}$ and 1250 $cm^{-1}$ some weak features coincide with water and methane lines. These coincidences may point to errors in the spectroscopy databases. However, they may also
5   point to broadband radiometric biases, such as atmospheric continuum contributions or non-linear instrumental effects. The





impact of such effects on the spectrum is a function of the total optical density, explaining the different size of the spectral features in the component below and above $1250\,\mathrm{cm}^{-1}$.

## 4.1 Bias-corrected methane retrievals

The first step of assessing the radiometric bias comprises the subtraction of the averaged residual from every GOSAT measurement before conducting a retrieval. In addition, we modify the forward model $\boldsymbol{F}$ by

$$\tilde{\boldsymbol{F}}(\boldsymbol{x}) = \boldsymbol{F}(\boldsymbol{x}) + \sum_i a_i \boldsymbol{p}_i \qquad (16)$$

adding principal components $\boldsymbol{p}_i$ with the amplitudes $a_i$ to be determined by the retrieval. Obviously, every addition of a principal component improves the spectral fit quality indicated by smaller $\chi^2$ values, but on the other hand, it increases noise propagation and instability of the retrieval. Therefore, a trade-off needs to be made between bias-mitigation and reduced precision. Adding the first principal component to $\tilde{\boldsymbol{F}}$, improves the overall shape of the inferred methane profile, lowering the overall bias. The noise propagation, on the other hand, is only slightly increased with respect to the retrievals without this retrieved scaling parameter. Accounting for additional principal components leads hardly to any improvement in the bias but does increase the standard deviation in the differences between retrieved and reference methane profiles and is henceforward not considered in this study.

The effect of including this bias-correction scheme in the retrieval algorithm on the retrieved methane profile is depicted in Figure 9. On the left in this figure the profiles from bias-corrected GOSAT-TIR measurements are depicted for different scenes with the scaled MACC profile as a reference. It is noted that only the MACC profile for the daytime land case is depicted as the profiles for the other scenes are very similar and have been left out for clarity. On the right the averaged difference between GOSAT and MACC is depicted and it clearly the bias in the profiles is almost fully corrected for. The bias is within 2% over the whole altitude range. Also the spread in the ensemble, given by the error bars is lower than in the non-corrected case (from $\approx 0.10\,\mathrm{ppmv}$ to $\approx 0.08\,\mathrm{ppmv}$ at $9\,\mathrm{km}$). In addition, the different retrieval performances for daytime and nighttime measurements, observations over land and ocean have been reduced and the daytime measurements over land are in line with the other three types of measurements.

For the other TCCON stations similar behaviour is found as can be seen in Fig. 10. For the retrieved methane concentration at $9\,\mathrm{km}$ altitude, the mean bias is -0.08% and the $1\sigma$ station-to-station variation in the bias is 0.76%. At this altitude, the discrepancy between daytime over-land scenes and the other three scenes is small (mean biases are respectively -0.31% and -0.01%; station-to-station variations are 0.83% and 0.72%). For $2\,\mathrm{km}$ altitude, we find that daytime over-land measurements show a systematic positive bias over all TCCON stations (mean bias is 0.97% with a station-to-station variation of 0.53%), whereas for the nighttime and ocean measurements, the corresponding biases are much smaller (mean of 0.07% and a variation of 0.16%). The daytime over-land biases may be explained by the fact that the HIPPO measurements are predominantly performed over the Pacific, and the few over-land measurements are not sufficiently different to fully account for the variability in the spectral residuals of all different scenes. Therefore, it may be that the correction is most applicable for scenes with a



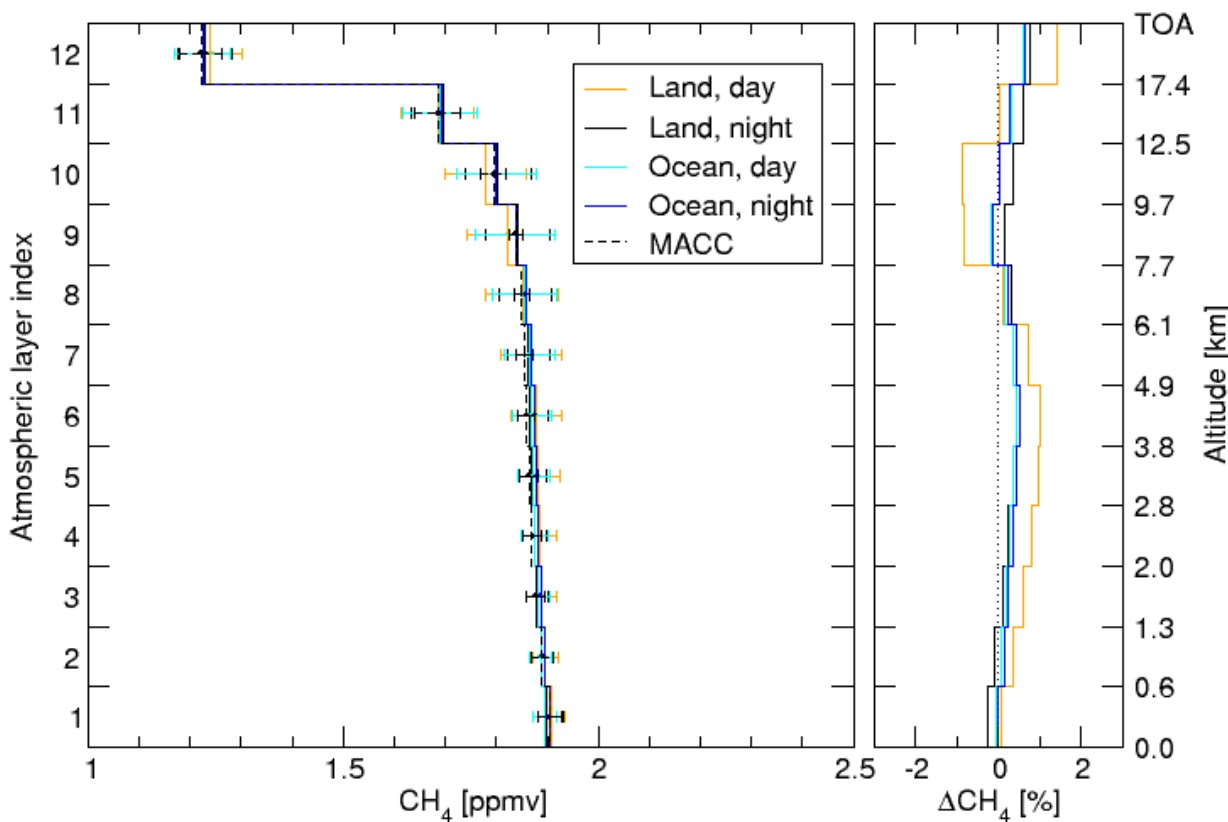

**Figure 9.** (left) Retrieved methane profiles from GOSAT-TIR measurements over TCCON station Bialystok with the bias-correction scheme included in the retrieval algorithm. The data are divided for different scenes; land and ocean scenes during daytime (respectively orange and cyan) and nighttime (respectively black and blue). The dashed lines refer to the MACC profiles for the daytime land scenes, and are very similar to the profiles for the other 3 scenes and have been left out for clarity. (right) The relative difference between the GOSAT-TIR retrievals and the MACC profiles.




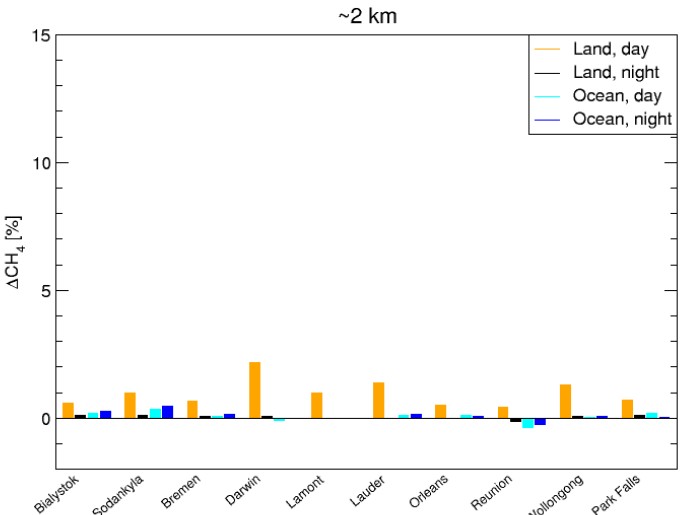

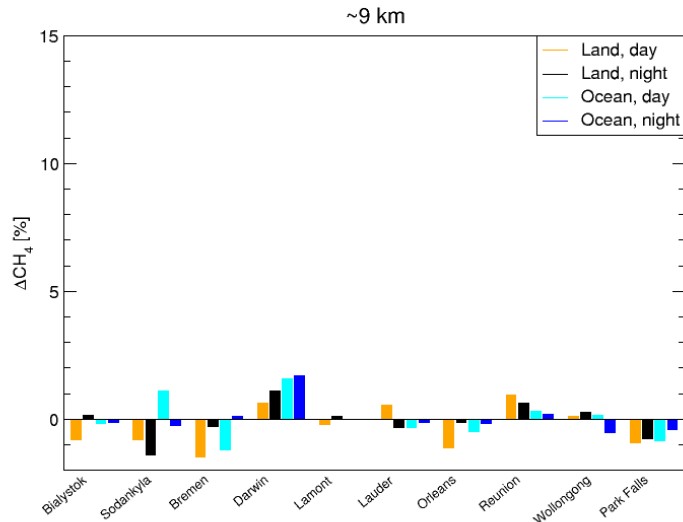

**Figure 10.** Bar-graphs for the relative deviation in the partial methane column at ≈ 2 km altitude (left) and ≈ 9 km (right) for ten different TCCON stations. The data are divided for different scenes; land and ocean scenes during daytime (respectively orange and cyan) and nighttime (respectively black and blue).





**Table 1.** Total methane column retrieval results after applying the bias-correction scheme for scenes over land during daytime. The first three columns pertain to the cloud filter based on fit parameters of GOSAT-TIR retrievals, whereas the last three columns refer to the filtering exploiting spectral features in GOSAT SWIR and NIR spectra. The average remaining bias is given for each TCCON station used in the current study along with the $1\sigma$ spread and the number of measurements passing the particular filter. The last two rows show the values for the whole ensemble.

| Station | TIR filtered | | | SWIR filtered | | |
|---|---|---|---|---|---|---|
| | $b$ | $\sigma$ | $n$ | $b$ | $\sigma$ | $n$ |
| Bialystok | 0.4 | 2.2 | 3370 | 0.8 | 1.4 | 355 |
| Sodankyla | 0.7 | 1.7 | 486 | 0.9 | 0.8 | 9 |
| Bremen | 0.1 | 2.1 | 1366 | 0.2 | 1.2 | 212 |
| Darwin | 1.3 | 2.1 | 1542 | 0.5 | 1.1 | 463 |
| Lamont | 0.6 | 2.1 | 6985 | 0.4 | 1.2 | 2307 |
| Lauder | 1.3 | 3.1 | 187 | 0.4 | 1.8 | 27 |
| Orleans | 0.1 | 1.9 | 3023 | 0.4 | 1.1 | 384 |
| Reunion | 0.7 | 1.8 | 194 | 0.9 | 1.1 | 48 |
| Wollongong | 0.9 | 2.4 | 2121 | 0.6 | 1.3 | 407 |
| Park Falls | 0.3 | 2.2 | 4367 | 0.6 | 1.3 | 947 |
| | | | | | | |
| $\bar{b}$ | 0.6 | 2.2 | | 0.6 | 1.2 | |
| $\sigma_b$ | 0.4 | - | | 0.2 | - | |

low thermal contrast. In the future, this shortcoming of our bias correction can be improved upon by an extended ensemble of airborne measurements, including over-land $CH_4$ and $N_2O$ measurements.

After establishing the bias correction, we finally consider the efficiency of the TIR cloud filter as discussed in Sec. 3.4. For GOSAT daytime over-land measurements, we compare the efficiency of the TIR cloud filter with that of the RemoTeC cloud clearing for the SWIR retrievals. Table 1 displays the average bias $b$ and its standard deviation $\sigma$ for GOSAT-TIR retrievals applying the two different cloud filters to the data. From the table it can be seen that the number of scenes $n$ passing the TIR filter is significantly higher than for the SWIR filter. The average results are consistent with both filtering methods as the mean bias of all stations is $\bar{b} = +0.6\%$ for both cloud filters. However, the station-to-station scatter in the bias $\sigma_b$, defined as the standard deviation of the mean biases per station, is $0.4\%$ and $0.2\%$ for the TIR and SWIR cloud filtered data, respectively. Also the scatter in the data are significantly lower in case of the SWIR cloud filter ($1.2\%$) compared to the TIR filter ($2.2\%$). It is noted that constraining the TIR filter criteria more stringently does not lead to a reduced scatter.



For the daytime land retrievals one may therefore consider to apply the SWIR filtering. However, this is not possible for ocean and nighttime observations. For consistency reasons we only consider TIR cloud filtering for all observations in this study.

We conclude that the cloud filter using SWIR spectral features is more able to filter GOSAT observations with respect to cloudiness than the TIR data filtering. However, on average both cloud filters are consistent and the cloud filter using TIR data does not introduce additional biases in the methane product.

## 5 Conclusions

Methane profile retrievals generally result in a positive bias when retrieved from thermal infrared spectra. In case of GOSAT TIR, this bias is 4–5% in the total methane column, and can amount to 10% at altitudes where the sensitivity peaks (typically 9 km). To account for this bias, a correction scheme has been developed. It has been shown that a simple additive or multiplicative scheme may result in a sufficiently accurate total methane column product, but that such schemes are insufficient to account for nonphysical structures in the retrieved profiles. In fact, these structures only yield correct total columns when they properly cancel. Especially in cases with enhanced methane abundances, in particular close to the surface or at high altitudes, this presumption may not be valid. In view of inversion schemes to determine methane sources and sinks, it is these scenes with enhanced methane that are most interesting, but may lead to erroneous values. Moreover, land-ocean transitions and differences between day and night are also not fully corrected for with these simple correction schemes in the case of GOSAT-TIR data.

In this study, we have developed a more elaborate bias correction scheme to account for all these aspects in methane retrievals from GOSAT-TIR spectra. The scheme is rooted in a principal component analysis of the spectral residuals between measurement and a forward model run with the best possible knowledge of the state of the atmosphere. Pivotal in this knowledge are $CH_4$ and $N_2O$ profiles which have been derived from HIPPO air campaign data. It has been shown that accounting for the average spectral residual and including one additional fitting parameter to scale the first principal component is sufficient to account for the bias within 2% when compared to the MACC methane fields (scaled to TCCON total columns). This is true for the whole altitude range from ground level to the top of the atmosphere and over all ten TCCON stations considered in this study. Moreover, the retrieval results from measurements over the ocean and the nighttime measurements over land, are all consistent with each other. Only at low altitudes, where the measurements have only limited sensitivity, the daytime measurements over land seem to show a persistent positive bias of $\approx 1\%$ at low altitudes. These scenes generally show a larger contrast between the Earth's skin temperature and the temperature of the lowest atmospheric levels, with respect to ocean scenes or nighttime observations. The reason that the bias correction scheme does not fully account for this bias in methane, may lie in the fact that the HIPPO campaigns are mostly performed over the Pacific, and the daytime land measurements may therefore be under-represented in the data set of residuals to be accounted for in a principal component analysis.

Nevertheless, the average bias in the retrieved GOSAT-TIR methane profile is less than 2% over the full altitude range, for all scenes over all TCCON stations, during day and night, when compared with MACC/TCCON values.



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
