# Peer review of "Methane profiles from GOSAT thermal infrared spectra"

_Atmospheric Measurement Techniques, 2017_

## Referee Comment (RC1) · Anonymous Referee #1 · 20 Jul 2017

This paper presents and evaluates a new $CH_4$ retrieval from GOSAT, using the thermal infrared channels between 1210-1310 cm-1 and a new cloud-screening procedure. The retrievals are evaluated by comparison with model profiles scaled to match TCCON retrievals of total $CH_4$ columns. A principal component analysis (PCA) bias correction scheme is also developed using airborne observations of $CH_4$ profiles from HIPPO II and III. The post bias-correction retrieval is shown to have baises of less than 2% over the entire altitude range and that the land-ocean and day-night differences in the performance of the initial retrievals are reduced or eliminated.

This is an interesting paper on a novel retrieval approach for the GOSAT instrument, and thus I think it should be accepted for publication after correction to address my minor issues listed below.

[Figure]

Minor Comments

P1, L6: Make clear the MACC model profiles are scaled to match the TCCON observations. i.e. "scaled to the TCCON total column". The phrasing on P2, L30-33 is more clear.

P1, L20-21: Some wildfires are also "natural" sources of CH4, so this discussion should be reworded.

P1, L25-26 and P2, L3: Please provide a few references for these statements – I agree with them, you just need references to support them in the text.

P2, L4-9: Also include the Cross-track Infrared Sounder aboard the Suomi-NPP satellite.

P2, L35: I don't think you need to note that the correction scheme helps here, as if it didn't you probably wouldn't be publishing it.

P3, L6-19: I would appreciate more discussion of GOSAT in the main text of the paper, specifically on the spectral resolution of the infrared band and on any known instrument or retrieval issues with the thermal infrared observations.

P4, L14: What version of the MT_CKD continuum did you use? The reference discusses several different versions.

P4, L20-22: Can you provide more details on how the actual line-by-line calculation is done?

P8, L10: This is not true as stated, as you have just shown in Figure 1 that the sensitivity of the methane retrieval varies quite a bit with altitude. I think you are trying to say that, after the averaging kernel of GOSAT is applied to the MACC/TCCON columns, they have similar sensitivity and thu can be compared? If so, that is not currently clear in the text.

P9, L16-19: I'm not convinced that this cloud-clearing algorithm is sufficient for the

type of validation study you are doing here. Did you make any independent checks to confirm that the cloud-filtered profiles were likely cloud free, say using independent observations from other bands for the daytime cases? How large of a cloud AOD can your procedure miss?

P9, L21: Is the chi-squared check considered part of the cloud filter?

Typos and Style Suggestions

P1, L18 and L20: I'd say "the year 1750" in both places, as the first time I read this I thought you were saying this was the pre-industrial concentration of CH4 in ppbv.

P2, L24: Check the format of these references.

P7, L18: typo in "A priori"

P9, L12: I'm not sure what "we pertain to" means here, I think you mean something like "we focus on"

P11, L6: This is a bar chart, not a histogram.

---

## Referee Comment (RC2) · V. Payne (Referee) · 19 Aug 2017

This paper describes an approach for retrieval and validation of methane profiles from GOSAT thermal infrared radiances. The authors develop a correction to the radiances, based on radiance closure between GOSAT measured radiances and forward-modeled radiances using HIPPO aircraft profile measurements as input. Using a separate validation set, the authors show that this correction significantly mitigates the high bias in upper tropospheric methane retrievals observed in this work. (As this paper points out, a similar bias has also been observed in other nadir thermal infrared methane retrievals.)

In my mind, there are two particular strong points of interest in this paper. One is

the correction approach, which could be readily applied to methane (and potentially to other retrievals of well-mixed gases) from other nadir-sounding thermal infrared instruments, such as TES, AIRS, CrIS and IASI. The authors may be interested to know that a similar approach, using empirical orthogonal functions is currently being applied within the OCO-2 Level 2 algorithm (see comments below). The other point of interest is the use of model profiles, scaled to TCCON total column estimates, for validation. This approach allows proper consideration of the vertical sensitivity of the thermal infrared retrievals in the comparison, while providing a relatively large number of samples for validation, with samples that are independent of the measurements used to develop the correction. The scope is well-suited to AMT, the work presented represents an interesting contribution to the field and the paper should be published after minor revisions. Minor comments, suggestions and questions for the authors are listed below.

In general: How good are the MACC CH4 profiles? Can the authors provide any references to model validation? Are the profile comparisons sensitive to uncertainties in the model representation of the stratosphere?

Abstract, lines 14-15: "This filter....is consistent with the cloud filter based on the GOSAT-SWIR measurements, despite the fact that the TIR-filter is less stringent". I was not clear on what this means. When you say the filter is consistent, do you mean that the bias in the retrieved profile does not change according too which filter is used? Consider changing the wording to say that the bias (rather than the filter) is consistent?

Page 2, lines 1-12: Please also list the Cross-track Infrared Sounder (CrIS). There is a CrIS flying on the Suomi-NPP satellite, launched in 2011, and there will be follow-on instruments on the JPSS satellite series. I am not aware of a publication on CrIS CH4 retrievals to date, but there are definitely people working on those.

Page 2, lines 23-29: This discussion of previous work is a little hard to follow and would benefit from some re-wording for clarification of various points. Papers from Saitoh (2012) and Holl (2016) cannot both present "first results". It would be helpful to

clarify that the Saitoh (2012), Holl (2016) and Zhou (2016) papers all discuss GOSAT TIR results from the same algorithm, and that algorithm is different from the one that you are using here. (Degree of signal should be degrees of freedom for signal?) When you say that the degrees of freedom for signal are significantly lower than 1, are you referring to the degrees of freedom for signal in that other algorithm? This was not totally clear from the text.

Zhou et al. (2016) compare results from AIRS (not IASI) and GOSAT. The statement about "A prevalent bias. . .between both satellite retrievals" is confusing. Since the Zhou et al. paper does not appear to include any independent validation measurements and deals only with a comparison between two satellite retrievals, I assume that you are referring purely to the difference between the retrievals, in which case, you should state clearly which one is biased high relative to the other.

As an aside, a point that is not discussed in the Zhou et al. paper, but which has been referred to in AIRS papers and presentations (for example, in Xiong et al. [2008]) is that in those AIRS CH4 retrievals, absorption coefficients are tuned within the radiative transfer algorithm in order to produce better agreement with validation data (a different form of correction).

Xiong, X., C. Barnet, E. Maddy, C. Sweeney, X. Liu, L. Zhou, and M. Goldberg (2008), Characterization and validation of methane products from the Atmospheric Infrared Sounder (AIRS), J. Geophys. Res., 113, G00A01, doi:10.1029/2007JG000500.

Page 2, lines 34-36: I think it would be good to refer here to the use of a similar correction approach using empirical orthogonal functions within the OCO-2 Level 2 algorithm. To my knowledge, the OCO-2 approach is not discussed in any journal papers to date, but you can find discussion of the use of empirical orthogonal functions in the OCO-2 Algorithm Theoretical Baseline Document (ATBD), available at: https://docserver.gesdisc.eosdis.nasa.gov/public/project/OCO/OCO2_L2_ATBD.V6.pdf

Page 3, line 27: It is not clear to me what is meant by "an effective H2O column to

calculate the water continuum independently from the water vapor absorption lines." Can you please expand on this point?

Page 6, Fig 1: Please label the altitude axis on the right hand side of the figure.

Page 7, line 1: Suggest removing the word "reduced".

Page 7, lines 18-19: "the fact that the null space contribution of the integrated methane column is typically in the order of 30 %". Did you show this somewhere? Please elaborate.

Page 9, line 7: Suggest replacing "cloud clearing" with "cloud screening", since the term cloud clearing has a particular meaning to some members in the TIR sounding community (Susskind et al., 2003).

Susskind et al., IEEE TRANSACTIONS ON GEOSCIENCE AND REMOTE SENSING, VOL. 41, NO. 2, FEBRUARY 2003

Page 10, line 9: "Indicatory" is not a word that would be commonly used. Suggest saying instead that the difference is representative.

Page 11, Figure 4: Why choose this order for the TCCON stations? Consider arranging them by latitude.

Grammar/typographical errors:

Page 3, line 14: Suggest splitting the points about the 10 km footprint and the sparse spatial sampling into two separate sentences for clarity.

Page 3, line 15: Coarse should be course.

Page 3, line 16: Should this be v160160?

Page 2, line 15: Tropospherical should be tropospheric.

---

## Author Comment (AC1) · 20 Oct 2017

**Methane profiles from GOSAT thermal infrared spectra**

Arno de Lange[1] and Jochen Landgraf[1]

[1]SRON Netherlands Institute for Space Research, Utrecht, The Netherlands

*Correspondence to:* Arno de Lange (A.de.Lange@sron.nl)

**Abstract.**

**1 Introduction**

**General**

We would like to thank both reviewers for the constructive comments that aided us to improve our manuscript. In this document we provide our replies to the reviewer's comments. The original comments made by the reviewer are typeset in italic font. Following every comment we give our reply. We provide a new version of the manuscript but in our replies to the comments we provide line numbers, page numbers and figure numbers referring to the original version of the manuscript, if not stated differently.

**Response to Referee #1**

**Minor Comment 1**

*P1, L6: Make clear the MACC model profiles are scaled to match the TCCON observations. i.e. "scaled to the TCCON total column". The phrasing on P2, L30-33 is more clear.*
  Changes: Wording of P1, L6 is now in line with phrasing on P2, L30-33.

**Minor Comment 2**

*P1, L20-21: Some wildfires are also "natural" sources of CH4, so this discussion should be reworded.*
  Changes: Clear distinction is made between natural and anthropogenic sources (P1, L20-21).

**Minor Comment 3**

*P1, L25-26 and P2, L3: Please provide a few references for these statements – I agree with them, you just need references to support them in the text.*
  Changes: Two references have been added to support both statements (P1, L25-26 and P2, L3).

**Minor Comment 4**

*P2, L4-9: Also include the Cross-track Infrared Sounder aboard the Suomi-NPP satellite.*

Changes: At the end of P2, L9, CRiS aboard Suomi-NPP has been added to the listed satellite instruments.

**Minor Comment 5**

*P2, L35: I don't think you need to note that the correction scheme helps here, as if it didn't you probably wouldn't be publishing it.*

Changes: P2, L35-P3, L2 have been re-worded.

As a stand-alone sentence the referee would be right. However, in the article this particular sentence is used as a bridge to the following line. To prevent confusion, the two sentences have been re-worded.

**Minor Comment 6**

*P3, L6-19: I would appreciate more discussion of GOSAT in the main text of the paper, specifically on the spectral resolution of the infrared band and on any known instrument or retrieval issues with the thermal infrared observations.*

Changes: We have added the spectral resolutions of the bands (P3, L11). Regarding instrumental issues, there is the concern that a small non-linearity is unaccounted for in the L1b version used in this study, which is also added to the text (P3, L18).

We are not aware of specific GOSAT retrieval issues. There is, however, a general observation of a (positive) bias in methane TIR retrievals that is not fully understood, as already mentioned in the text (P2, L13-22).

**Minor Comment 7**

*P4, L14: What version of the MT_CKD continuum did you use? The reference discusses several different versions.*

Changes: Version MT_CKD_2.5 has been added (P4, L14)

**Minor Comment 8**

*P4, L20-22: Can you provide more details on how the actual line-by-line calculation is done?*

Changes: The line-by-line calculation follows the RemoTeC implementation and this remark has been added to the text with the proper reference (P4, L19).

**Minor Comment 9**

*P8, L10: This is not true as stated, as you have just shown in Figure 1 that the sensitivity of the methane retrieval varies quite a bit with altitude. I think you are trying to say that, after the averaging kernel of GOSAT is applied to the MACC/TCCON columns, they have similar sensitivity and thu can be compared? If so, that is not currently clear in the text.*

Changes: For clarity SWIR is now explicitedly mentioned in P8, L8.

I believe there is a misunderstanding here. Figure 1 refers to TIR averaging kernels, but the statement (P8, L10) refers to SWIR retrievals and Figure 1 is therefore not relevant in this discussion (nor mentioned in the text). For the SWIR retrievals the statement is, to a very high degree, true. To avoid confusion SWIR is now explicitely mentioned in the text.

**Minor Comment 10**

*P9, L16-19: I'm not convinced that this cloud-clearing algorithm is sufficient for the type of validation study you are doing here. Did you make any independent checks to confirm that the cloud-filtered profiles were likely cloud free, say using independent observations from other bands for the daytime cases? How large of a cloud AOD can your procedure miss?*

No changes.

In this study we actually used the method proposed by the referee to check the cloud filter against the independent observations from the SWIR bands (P20, L3-11) and the quality of the filter is sumarised in Table 1 on P20.

The suggestion by the referee to identify the detection limit in terms of cloud optical depth, sounds reasonable and is actually a quantity provided by the SWIR filter. However, there are two reasons for not including this in the current study: First, the cloud optical depth from the SWIR filter is not a validated product, but merely a filter quantity, and secondly, the cloud optical depth alone is of limited use only since the vertical sensitivity of the TIR retrievals. For instance, a thick low cloud may be missed by the TIR filter, without impacting the retrievals because of the loss of sensitivity towards the surface.

We have shown that the performance of the TIR and SWIR filters are almost identical in terms of biases (Table 1, P20), but the spread is larger in case for the TIR cloud filter (Table 1, P20). Therefore we believe that this (admittedly crude) cloud filter does not introduce additional biases (which is the focus of the current study) and we propose to leave the text as is.

**Minor Comment 11**

*P9, L21: Is the chi-squared check considered part of the cloud filter?*

No changes.

Yes, the chi-squared check is indeed considered part of the cloud filter, which is in correspondence with the cloud filter from the SWIR bands. This test identifies the failure of the forward model to capture all spectral features of the observation. The forward model does not account for clouds, and a large chi-squared may therefore indicate the occurence of clouds in the observation.

**Typos and Style Suggestions 1**

*P1, L18 and L20: I'd say "the year 1750" in both places, as the first time I read this I thought you were saying this was the pre-industrial concentration of CH4 in ppbv.*

Changes: As per suggestion (P1, L18 and L20).

**Typos and Style Suggestions 2**

*P2, L24: Check the format of these references.*
  Changes: The references are now correct (P2, L24).

**Typos and Style Suggestions 3**

5 *P7, L18: typo in "A priori"*
  Changes: As per suggestion (P7, L18).

**Typos and Style Suggestions 4**

*P9, L12: I'm not sure what "we pertain to" means here, I think you mean something like "we focus on"*
  Changes: As per suggestion (P9, L12).

10 **Typos and Style Suggestions 5**

*P11, L6: This is a bar chart, not a histogram.*
  Changes: As per suggestion (P11, L6).

**Response to V. Payne (Referee #2)**

**Comment 1**

15 *In general: How good are the MACC CH4 profiles? Can the authors provide any references to model validation? Are the profile comparisons sensitive to uncertainties in the model representation of the stratosphere?*
  Changes: A second verification study has been added to the text (P10, L1)
  The quality of the MACC fields are being discussed in lines P9,L29-P10,L3. In these lines we refer to a study verifying that the model delivers methane fields within 1% uncertainty, albeit with different input data as used in the current setup

20 (NOAA-ESRL as opposed to GOSAT-SWIR). For completeness, we have now added a second reference in which the MACC fields (based on SCIAMACHY data) are verified to be again within 1% with independent observations. We have estimated the uncertainty in the current setup to be 2%, which seems therefore to be on the safe side.

**Comment 2**

*Abstract, lines 14-15: "This filter. . ..is consistent with the cloud filter based on the GOSAT-SWIR measurements, despite the*
25 *fact that the TIR-filter is less stringent". I was not clear on what this means. When you say the filter is consistent, do you mean that the bias in the retrieved profile does not change according too which filter is used? Consider changing the wording to say that the bias (rather than the filter) is consistent?*

Changes: Abstract L14-15 have been adapted to clearly state that a) no additional biases are introduced by the TIR-filter (wrt. SWIR) and b) the acceptance rate of observations is higher for the TIR-filter but the uncertainty as well (wrt. SWIR)

**Comment 3**

*Page 2, lines 1-12: Please also list the Cross-track Infrared Sounder (CrIS). There is a CrIS flying on the Suomi-NPP satellite, launched in 2011, and there will be follow-on instruments on the JPSS satellite series. I am not aware of a publication on CrIS CH4 retrievals to date, but there are definitely people working on those.*

Changes: At the end of P2, L9, CRiS aboard Suomi-NPP has been added to the listed satellite instruments.

**Comment 4**

*Page 2, lines 23-29: This discussion of previous work is a little hard to follow and would benefit from some re-wording for clarification of various points. Papers from Saitoh (2012) and Holl (2016) cannot both present "first results". It would be helpful to clarify that the Saitoh (2012), Holl (2016) and Zhou (2016) papers all discuss GOSAT TIR results from the same algorithm, and that algorithm is different from the one that you are using here.*

Changes: "First" has been omitted (P2, L24). The statements that these papers stem from the same algorithm and we use a different one, are included (P2, L30).

**Comment 5**

*(Degree of signal should be degrees of freedom for signal?) When you say that the degrees of freedom for signal are significantly lower than 1, are you referring to the degrees of freedom for signal in that other algorithm? This was not totally clear from the text.*

Changes: It should indeed be "degrees of freedom for signal" and has been adapted (P2, L26). Two other occurences in the text have been adapted as well (P2, L30 and P6, L3). We were indeed referring to the other algorithms and this has now been made explicit (P2, L26).

**Comment 6**

*Zhou et al. (2016) compare results from AIRS (not IASI) and GOSAT. The statement about "A prevalent bias. . .between both satellite retrievals" is confusing. Since the Zhou et al. paper does not appear to include any independent validation measurements and deals only with a comparison between two satellite retrievals, I assume that you are referring purely to the difference between the retrievals, in which case, you should state clearly which one is biased high relative to the other. As an aside, a point that is not discussed in the Zhou et al. paper, but which has been referred to in AIRS papers and presentations (for example, in Xiong et al. [2008]) is that in those AIRS CH4 retrievals, absorption coefficients are tuned within the radiative transfer algorithm in order to produce better agreement with validation data (a different form of correction). Xiong, X., C.*

*Barnet, E. Maddy, C. Sweeney, X. Liu, L. Zhou, and M. Goldberg (2008), Characterization and validation of methane products from the Atmospheric Infrared Sounder (AIRS), J. Geophys. Res., 113, G00A01, doi:10.1029/2007JG000500.*

Changes: The confusion between IASI and AIRS has been cleared (P2, L28). It is now explicitedly stated that AIRS is biased high wrt. GOAST (P2, L27-29). The bias correction approach in the AIRS retrievals has been added to the discussion 5 (P2, L14).

**Comment 7**

*Page 2, lines 34-36: I think it would be good to refer here to the use of a similar correction approach using empirical orthogonal functions within the OCO-2 Level 2 algorithm. To my knowledge, the OCO-2 approach is not discussed in any journal papers to date, but you can find discussion of the use of empirical orthogonal functions in the OCO-2 Algorithm Theoretical Baseline* 10 *Document (ATBD), available at: https://docserver.gesdisc.eosdis.nasa.gov/public/project/OCO/OCO2_L2_ATBD.V6.pdf*

Changes: We agree that the approach by the OCO-2 team is very similar and we have added this to the text (P3, L2), including the reference.

**Comment 8**

*Page 3, line 27: It is not clear to me what is meant by "an effective H2O column to calculate the water continuum independently* 15 *from the water vapor absorption lines." Can you please expand on this point?*

Changes: P3, L27 has been re-worded.

It turned out that a single water retrieval parameter was not sufficient to capture the spectral features of both the water vapour absorption lines and the continuum contribution. Therefore, two parameters are in the state vector incorporated to account for both contributions respectively. The text has been expanded to make this point clear.

20 ## Comment 9

*Page 6, Fig 1: Please label the altitude axis on the right hand side of the figure.*

Changes: Fig 1 (P6) has been changed. For similar reasons Figs 2 (P8), 5 (P12), and 6 (P13) have been changed as well.

**Comment 10**

*Page 7, line 1: Suggest removing the word "reduced".*

25 Changes: As per suggestion (P7, L1).

**Comment 11**

*Page 7, lines 18-19: "the fact that the null space contribution of the integrated methane column is typically in the order of 30 %". Did you show this somewhere? Please elaborate.*

Changes: An explicit explanation has been added to the text (P7, L18) with a reference to Figure 2 (P8).

**Comment 12**

*Page 9, line 7: Suggest replacing "cloud clearing" with "cloud screening", since the term cloud clearing has a particular meaning to some members in the TIR sounding community (Susskind et al., 2003). Susskind et al., IEEE TRANSACTIONS ON GEOSCIENCE AND REMOTE SENSING, VOL. 41, NO. 2, FEBRUARY 2003*

Changes: As per suggestion (P9, L17). An other occurence has been updated as well (P20, L5).

**Comment 13**

*Page 10, line 9: "Indicatory" is not a word that would be commonly used. Suggest saying instead that the difference is representative.*

Changes: As per suggestion (P10, L9).

**Comment 14**

*Page 11, Figure 4: Why choose this order for the TCCON stations? Consider arranging them by latitude.*

No changes.

It is the processing order and we did not put much thought into the arrangment of the figures. Rearranging them would only make sense when the order is rearranged in all figures, charts and tables. However, this needs to be done manually and is therefore error prone. We would like to request if keeping the current, admittedly peculiar, order is acceptable, as it will not lead to different conclusions or insights.

**Grammar/typographical errors 1**

*Page 3, line 14: Suggest splitting the points about the 10 km footprint and the sparse spatial sampling into two separate sentences for clarity.*

Changes: As per suggestion (P3, L12-14).

**Grammar/typographical errors 2**

*Page 3, line 15: Coarse should be course.*

Changes: As per suggestion (P3, L15).

**Grammar/typographical errors 3**

*Page 3, line 16: Should this be v160160?*

No changes.

During the reprocessing of the data under v160160, a small update was incorporated leading to v161160. It was decided to only process the remaining data under v161160 and keep the already processed data under v160160. Therefore the version is generally indicated as v16x160.

**Grammar/typographical errors 4**

*Page 2, line 15: Tropospherical should be tropospheric.*

Changes: As per suggestion (P2, L15).

**2 Conclusions**

**References**

---

## Author Response (AR2)

**Methane profiles from GOSAT thermal infrared spectra**

Arno de Lange[1] and Jochen Landgraf[1]

[1]SRON Netherlands Institute for Space Research, Utrecht, The Netherlands

*Correspondence to:* Arno de Lange (A.de.Lange@sron.nl)

**Abstract.**

**1 Introduction**

**General**

We would like to thank the Editor for the two constructive comments. Below we will provide our replies both comments. The original comments made by the Editor are typeset in italic font, followed by our reply. We provide a new version of the manuscript but in our replies to the comments we provide line numbers, page numbers and figure numbers referring to the original version of the manuscript.

**Response to Editor**

**Editor's Comment 1**

*During the original Editor response I had asked for the authors to discuss the uncertainties that result from comparing the TCCON total column to the thermal IR based total column. I think this was implicitly addressed in the manuscript by using the MACC model but these uncertainties should be explicitly estimated. You could use the approach discussed in Rodgers and Connor 2003: Rodgers, C. D. & Connor, B. J. Intercomparison of remote sounding instruments. Journal of Geophysical Research-Atmospheres 108, 4116 (2003).*

We fully agree with the Editor that the uncertainty by the MACC profiles to the total columns should be explicitedly calculated. However, this is not possible due to the lack of an estimate of the accuracy in these profiles. One of the major problems in satellite methane profile valdiation is that there is only a limited set of reference profiles of sufficient quality available. If these were available we would have validated directly with this reference data.

To answer your request more specifically, when following the suggested reference, one needs the error covariance matrix of the MACC data, and this is an unknown quantity. Even the diagonal elements of this matrix, which could be used when assuming no correlations between altitudes, are unavailable. The uncertainties in the MACC dataset is derived from two studies in which the total column of TM5 (the underlying model for the MACC methane data) is verified against independent measurments. Both studies are referred to in the text (p. 11, lines 1 and 5 respectively).

In order to map the uncertainty of the MACC methane data onto the uncertainty of the GOSAT-TIR methane total column, the averaging kernel should be applied to the uncertainty of the MACC methane profile. However, from the studies mentioned above, an uncertainty has been derived for the total columns, and not the profile. As a first order approach, one could assume that the uncertainty in the MACC profile is evenly distributed over the profile. However, in case of correlated errors in the MACC profile, this assumption would lead to a too low error estimate. To account for this, we assumed an uncertainty of 2%, rather than 1% as the other studies indicated. With this assumption, the error contribution of the MACC profiles is estimated to be 0.6% of the overall 2% uncertainty in the GOSAT-TIR total column values (see p11).

**Editor's Comment 2**

*Also, please remove or modify this statement (Page 9 Line 14) "Here, both the ground-based and satellite observations show homogeneous methane retrieval sensitivity over all atmospheric altitudes, leading to highly accurate estimates of the total column of methane rather than a profile. " as the statement is mis-leading. The TIR based estimates are primarily sensitive to the mid-troposphere to lower stratosphere methane and NOT the total column.*

I am afraid that there is a misunderstanding. Indeed the statement would have been incorrect if it would refer to TIR measurements, but it refers to measurements in the SWIR. Since both referee Payne and the Editor are confused by the statement, we have adapted the wording to explicitly refer to SWIR measurements:

"Here, both the ground-based and satellite observations show homogeneous methane retrieval sensitivity ..." has been changed into "In this particular case of the SWIR wavelength regime, both the ground-based and satellite observations show homogeneous methane retrieval sensitivity ..."

**2 Conclusions**